# Leptospirosis and Coinfection: Should We Be Concerned?

**DOI:** 10.3390/ijerph18179411

**Published:** 2021-09-06

**Authors:** Asmalia Md-Lasim, Farah Shafawati Mohd-Taib, Mardani Abdul-Halim, Ahmad Mohiddin Mohd-Ngesom, Sheila Nathan, Shukor Md-Nor

**Affiliations:** 1Department of Biological Sciences and Biotechnology, Faculty of Science and Technology, Universiti Kebangsaan Malaysia, UKM, Bangi 43600, Selangor, Malaysia; asmaliaccb@gmail.com (A.M.-L.); sheila@ukm.edu.my (S.N.); shukor@ukm.edu.my (S.M.-N.); 2Herbal Medicine Research Centre (HMRC), Institute for Medical Research (IMR), National Institue of Health (NIH), Ministry of Health, Shah Alam 40170, Selangor, Malaysia; 3Biotechnology Research Institute, Universiti Malaysia Sabah, Jalan UMS, Kota Kinabalu 88400, Sabah, Malaysia; mardaniccb@gmail.com; 4Center for Toxicology and Health Risk, Faculty of Health Sciences, Universiti Kebangsaan Malaysia, Kuala Lumpur 50300, Federal Territory of Kuala Lumpur, Malaysia; aksmohiddin@yahoo.com

**Keywords:** coinfection, diagnostic, *Leptospira*, microbiome, transmission, pathogenic

## Abstract

Pathogenic *Leptospira* is the causative agent of leptospirosis, an emerging zoonotic disease affecting animals and humans worldwide. The risk of host infection following interaction with environmental sources depends on the ability of *Leptospira* to persist, survive, and infect the new host to continue the transmission chain. *Leptospira* may coexist with other pathogens, thus providing a suitable condition for the development of other pathogens, resulting in multi-pathogen infection in humans. Therefore, it is important to better understand the dynamics of transmission by these pathogens. We conducted Boolean searches of several databases, including Google Scholar, PubMed, SciELO, and ScienceDirect, to identify relevant published data on *Leptospira* and coinfection with other pathogenic bacteria. We review the role of the host-microbiota in determining the synanthropic interaction of *Leptospira* sp. with other bacteria, thus creating a suitable condition for the leptospira to survive and persist successfully. We also discuss the biotic and abiotic factors that amplify the viability of *Leptospira* in the environment. The coinfection of leptospira with pathogenic bacteria has rarely been reported, potentially contributing to a lack of awareness. Therefore, the occurrence of leptospirosis coinfection may complicate diagnosis, long-lasting examination, and mistreatment that could lead to mortality. Identifying the presence of leptospirosis with other bacteria through metagenomic analysis could reveal possible coinfection. In conclusion, the occurrence of leptospirosis with other diseases should be of concern and may depend on the success of the transmission and severity of individual infections. Medical practitioners may misdiagnose the presence of multiple infections and should be made aware of and receive adequate training on appropriate treatment for leptospirosis patients. Physicians could undertake a more targeted approach for leptospirosis diagnosis by considering other symptoms caused by the coinfected bacteria; thus, more specific treatment could be given.

## 1. Introduction

Leptospirosis is a zoonotic disease caused by the spirochete bacteria *Leptospira* spp. that belongs to the genus *Leptospira* [1]. There are 20 species of *Leptospira* categorised into 24 serogroups with more than 300 serovars [2]. It is estimated that there are one million apparent *Leptospira* infections annually, with nearly 60,000 deaths [3,4], rendering leptospirosis a significant threat to the global public [5].

The number of leptospirosis outbreaks is widespread throughout tropical and subtropical countries and frequently associated with natural disasters such as extreme flooding and heavy rainfall [6,7]. Direct contact with water, soil and vegetation contaminated with pathogenic leptospires during water-related incidence has been widely documented as the risk factor for leptospirosis [8,9,10]. Occupational activities that place individuals at high risk of infection include farming, agriculture, mining, sewage maintenance, and military action [11,12,13,14,15].

The transmission of *Leptospira* between humans, animals, and the environment is complex and not well documented [16]. Although the presence and survival capability of multiple strains of *Leptospira* in environmental samples may vary (soil, water, streams, and sewers) [17,18], the dynamics and mechanism of *Leptospira* in the environment are still unclear [19]. Although pathogenic leptospira can produce biofilm in nature [20,21,22], the interaction with other microorganisms in the environment might support the survival and persistence of leptospiral pathogens outside the host [23,24].

Coinfection can influence epidemiology and disease severity [25]. A coinfection involving simultaneous infection of several pathogens in the same host may result in different types of diseases [26]. A coinfection is synonymous with simultaneous infection, mixed infection, multiple infections, concomitant infection, concurrent infection, polyinfection, polyparasitism, and multiple parasitism of an individual host [27,28]. Coinfection involves infectious pathogens from various taxa levels (bacteria, fungi, parasites, and viruses) and genetic variations of the same infectious agents [29]. Coinfections are more likely to have detrimental impacts on host health compared to mono-infections. High pathogen abundance and simultaneous interaction could significantly affect infection dynamics by changing host vulnerability and the risk of multiple infections simultaneously.

For instance, leptospirosis co-infection with other diseases such as melioidosis [30], dengue fever [31], malaria [32], and typhus [33] has been well documented. Such co-infection typically leads to misdiagnosis, increases the severity of the disease, and can even benefit the causative agents [34]. The treatment of leptospirosis coinfection with other diseases may bring additional challenges if the symptom of the coinfection is similar to other common diseases [35,36,37]. Environmental exposure and the similar nature between pathogenic environmental bacteria may increase the possibility of coinfection and lead to miss or underdiagnosis. However, it is important to develop a better knowledge of the dynamics of transmission of these pathogenic *Leptospira* and ecological drivers of coinfections to improve a targeted public health response. This review aims to provide an overview of the present knowledge of leptospirosis and the probability of leptospirosis coinfection with environmental pathogenic bacteria. Given the lack of data on leptospirosis coinfection with environmental bacteria, we also include other potential coinfection.

## 2. *Leptospira*—An Overview of Epidemiology, Transmission and Persistence in the Environment 

*Leptospira*, derived from the Greek, term *leptos* (small) and the Latin word *speira* (a coil), is a single, motile, spiral-shaped, Gram-negative spirochete with internal flagella and finely coiled [38]. The bacteria are usually 0.1–0.2 µm in diameter and between 6 to 20 µm in length. Their helical amplitude is approximately 0.1 µm to 0.15 µm, with a wavelength of 0.5 µm. *Leptospira* demonstrates two forms of movement, translational and rotational [39]. Its morphology can differ when sub-cultured in vitro but restored when passaged in hamsters [40]. Leptospires are aerobic microbes with optimal growth between 28 °C and 30 °C. They also grow in simple media enriched with ammonium salts [41], vitamin B_1_, B_12_ [42], and long-chain fatty acids [43].

Leptospires are spirochaetes that belong to the order Spirochaetales, a family of Leptospiraceae. The Leptospiraceae family consists of two genera, *Leptospira* and *Leptonema*, and the genus *Leptospira* is divided into pathogenic, intermediate pathogenic, and non-pathogenic *Leptospira* [40]. Several species are classified within each category based on phenotypic properties and genotypic classification. The genus *Leptopsira* has been divided into three lineages based on pathogenicity and genetic analysis with pathogenic, saprophytic and intermediate pathogenic species (Table 1) [44,45,46,47,48]. Within each species, *Leptospira* is further subdivided into serovars, which is the smallest taxonomic unit of the bacteria. Approximately 300 pathogenic serovars have been identified, 88 of which belong to the well-characterised pathogenic species, *Leptospira interrogans* [36]. However, classification based on whole-genome sequencing may increase the number of the species in the future [48,49].

### 2.1. Epidemiology of Leptospira

Leptospirosis is distributed globally in rural and urban areas. The disease occurs in the temperate and tropical regions, with higher incidence rates in the tropics [50]. It remains one of the most important public health issues due to the wide range of serovars circulating within large reservoirs. Large populations of rodents, farm animals and dogs are capable of being colonised within the kidneys and shedding of *Lepospira* in urine which increases the spread to the environment [51,52,53]. If *Leptospira* that persist in the renal and urinary systems are shed from animals, the bacteria can survive under optimum environmental conditions for months and up to a year [54,55]. Animals that have recovered from leptospirosis may continue to be carriers of *Leptospira* within the renal tubules for several months [56].

Additionally, climate factors may influence *Leptospira* infectivity by providing ideal conditions for *Leptospira* to survive for prolonged periods of time, with transmissions becoming worse during heavy rainfall and flooding. *Leptospira* can easily contaminate the environment and pose a greater risk of infection through skin wounds during heavy rain. The water hits land surfaces and its either stored in the subsurface as soil moisture, or lost to evapotranspiration into freshwater bodies which leads to humans to be exposed [57,58,59]. This situation increases the possibility of Leptospira infections among rodents and increased interaction with humans will lead to increased occurrence of leptospirosis [60]. Reports on flooding is a significant factor for the increase in leptospirosis cases, and are common in Thailand, the Philippines, Sri Lanka, and Malaysia [61,62,63,64].

Other factors such as poverty, urbanisation, and rapid population growth may have an impact on the high prevalence of *Leptospira* infections in wildlife, domestic animals, and humans [65]. These situations increase reservoir populations and compromise basic health care services, sanitation, waste management, whilst exposing humans to *Leptospira*. The particular areas that overflow by heavy rainfalls that cannot be drained due to the faulty sanitation network, causing rodents to abandon their burrows and contaminate the water with their urine [66,67]. In addition to contaminated floodwater, rodents increase the possibility of transmitting bacteria to humans through cuts and abrasions of the skin [68], following prolonged immersion in contaminated water [51]. Transmission is most likely to occur during the cleaning of the flooded muddy houses and surroundings.

Occupational groups such as banana plantation farmers, sugarcane workers, paddy farmers, veterinarians, animal shelter employees, hunters, wet market workers and abattoir workers are at high risk of infection due to their job description involves contact with water and soils [69,70]. They are typically prone to skin cuts, abrasions during the activities in the particular areas which are providing a suitable environment and are rich with a source of food favouring the presence of rodents [71]. The rodent might share similar places with people, easily moving around and excreting bacteria into the environment via urine [72]. Farm labourers are especially vulnerable to injury, increasing their susceptibility to bacterial infections, partially due to a lack of adherence to the need for personal protective equipment.

### 2.2. Leptospira Persistence in the Environment

The ecological and environmental metabarcoding of DNA now routinely includes studies on the persistence and survival of *Leptospira* in the environment [73,74,75,76]. Using this approach, Sato et al. [75] discovered three pathogenic strains and non-pathogenic *Leptospira* strains from aquatic samples, which correlated with the level of precipitation. Further studies have also associated short to long term persistence in soil and water and the ability of the pathogen to survive and remain infectious outside the host [77]. Environmental parameters such as salinity, pH, temperature, humidity and high oxygen levels play a crucial role in the survival of *Leptospira* in the environment [38,78,79].

Many studies worldwide have shown a significant correlation between soil chemical properties and the survival of *Leptospira* bacteria. For example, in India, soil samples from rice fields and nearby stables were highly prevalent for *Leptospira* compared to urban sites [80]. Similarly, in Australian cane fields, the bacteria can survive up to seven weeks in acidic soil at pH 6.2 and three weeks in the rainwater-flooded soil [81]. Thibeaux et al. [48] reported that soil biochemical properties might determine the persistence of *Leptospira* strains and the risk of transmission to humans.

The presence of *Leptospira* is also prevalent in water bodies such as groundwater, rivers, sewage, and surface water in tropical and non-tropical regions [82]. The aqueous environment associated with salt concentration, high oxygen, pH, and alkalinity was also considered a key factor in the transmission of *Leptospira* to its host [83]. Optimal conditions for *Leptospira* growth have been reported at pH 6.7 to 7.3 in river water and dissolved solid salts surroundings [84]. However, a high concentration of salt is unsuitable for the long-term survival of pathogenic *Leptospira* [84], while other studies demonstrated the presence of *Leptospira* under neutral pH and slightly alkaline conditions (up to pH 8.0) [85]. Saito et al. [86] reported a steady decrease in pathogenic *Leptospira* motilities after 10 h of exposure. Hence, there is a critical need to address the survival of *Leptospira* in the environment with other factors such as biotic and abiotic interactions with other environmental pathogens leading to coinfections.

A study in Kelantan, Malaysia, identified pathogenic *Leptospira* from soil and water samples. Out of 42 samples of water and soil, 42.8% were positive for pathogenic *Leptospira* such as *Leptospira kmetyi* (17%), followed by intermediate strains of *Leptospira wolffi* (7%), *Leptospira licerasiea* (5%), *Leptospira fainei* (2%), and *Leptospira inadai* (2%). Culture-positive *Leptospira* obtained from the water samples was lower at 19.1%, likely due to toxic components within the sampled cloudy and foul-smelling water that limits *Leptospira* survivability [30].

## 3. Leptospira Coinfection, Is It Possible?

Currently, bacterial-viral coinfection is one of the biggest medical concerns, resulting in increased mortality rates is one of the biggest medical concerns. Approximately 30% of human diseases are caused by coinfection and could reach up to 80% in human communities [87]. Leptospirosis coinfection with other pathogenic bacteria in humans is a special case and also known as “the great mimicker” [35,88,89]. Unfortunately, no pathogenic effects and symptoms are observed in the patients at the early stage. The clinical presentations of leptospirosis coinfection are considerably overlapping, leading to misdiagnosis and mistreatment. Patients may develop fever, abdominal pain, myalgia, headache, vomitting, as well as long-term infections or death [35,36,90,91].

The majority of leptospirosis cases are mild and resolve spontaneously. Minor cases of leptospirosis resolve with time and oral antibiotics such as doxycycline, azithromycin, ampicillin and amoxicillin, which are administered based on the severity of the illness. Treatment with antibiotics must be initiated as soon as a provisional diagnosis of leptospirosis is suspected, regardless of the length of the symptom [92]. Furthermore, most of the coinfected patients respond effectively to fluid therapy, doxycycline and closely monitoring of their platelet count and hematocrit [93,94]. However, in some leptospirosis coinfections, treatment could not be assessed due to the lack of data, and it is critical to determine these different of infections that can provide relevant treatment decisions for patients with coinfections [95].

## 4. Leptospira Coinfection in Humans

It is often difficult to distinguish between the coinfection of *Leptospira* and other pathogens purely based on clinical presentation. Diagnosis is particularly challenging when broad spectrum of clinical phenotypes are presented [37]. *Leptospira* coinfection is more common with pathogenic agents causing dengue, malaria and scrub typhus although coinfection with arboviruses may also occur but is less frequently reported. During an outbreak of arboviruses-associated diseases, if conflicting microbiology or pathology results are obtained, clinicians should suspect a leptospiral coinfection, particularly among individuals from rural areas or travellers returning from epidemic areas [96]. The co-occurrence of leptospirosis with other diseases is summarised in Table 2.

### 4.1. Dengue

There have been increased reports of leptospirosis and dengue coinfection from several parts of the world especially Peru, Malaysia, and India [36,37,97,98]. The occurrence of leptospirosis and dengue coinfection has always been related to climate, with cases spiking during rainfall or monsoon seasons. During heavy rainfall, stagnant water is an ideal breeding ground for mosquitoes, while an increased rat population feeding on garbage brought to localities by flood waters transmits *leptospira* in their urine. Vella-Mendoza et al. [97] revealed that the prevalence of leptospirosis-dengue coinfection was higher in the age groups 20–44 and 45–59 at 37.5% and 25% respectively. Males are more vulnerable to coinfection, which could be due to a variety of factors such as work exposure, heavy rainfall and flooding.

Clinical features of coinfection may present as headache, myalgia and fatigue accompanying fever, making diagnosis more difficult, especially with an acute co-infection [37]. Diagnosis of leptospirosis includes detection of IgM antibodies by ELISA which is highly sensitive and IgM positive for dengue indicates a recent dengue infection. The polymerase chain reaction is also performed depending on the availability of the test. The majority of dengue treatments are driven by symptoms, whereas leptospirosis requires immediate administration of either penicillin, ceftriaxone or doxycycline to prevent complications.

**Table 2 ijerph-18-09411-t002:** Cases of leptopsira co-infections.

Location	Year	Study Type	No. Enrolled	No. Co-Infections	Co-Infection Prevalence (%)	Age	Diagnostic Test	References
**Dengue**								
Malaysia	2008	Case report	1	1	100	41	ELISA IgM	[99]
Malaysia	2012	Cross sectional study	84	32	38.1	Mean: 39.4	ELISA IgM, MAT	[35]
Puerto Rico	2012	Case report	1	1	100	42	ELISA IgM, PCR	[100]
Peru	2015	Case report	1	1	100	10	ELISA IgM, MAT	[101]
Sri Lanka	2015	Case report	1	1	100	52	ELISA IgM, IgG	[102]
Malaysia	2017	Retrospective study	268	11	4.1	30–32	ELISA IgM, MAT, PCR	[37]
Colombia	2018	Case study	1	1	100	87	ELISA IGM, PCR	[103]
South India	2018	Retrospective study	974	33	3.4	Mean: 37.4	ELISA IgM	[36]
Kelambakan	2018	Cross sectional study	100	4	4	21–30	ELISA IgM, PCR	[104]
Sri Lanka	2018	Retrospective study	6	1	16.7	17–73	ELISA IGM, PCR	[105]
India	NM	Case report	1	1	100	39	ELISA IgM, MAT, DFM	[106]
**Malaria**								
Thailand	1999–2002	Cross sectional study	18	7	39	20–38	ELISA IgM, MAT	[107]
India	2010	Case report	2	2	100	38 & 34	ELISA IgM, MAT	[108]
India	2011	Case report	18	18	100	28–40	ELISA IgM, SAT	[109]
Tamil Nadu	2012	Case report	220	48	22	Mean: 29	MSAT, MAT	[110]
India	2014	Case report	1	1	100	24	RMAT, MAT	[111]
Malaysia	2011–2014	Retrospective study	111	26	23.4	Mean: 33	MAT, BFMP	[112]
**Melioidosis**								
Malaysia	2010	Case report	20	4	20	29–60	PCR, Blood C & S	[113]
Malaysia	2010	Retrospective study	153	4	2.6	20–59	ELISA IgM, PCR	[114]
Malaysia	2016	Case report	1	1	100	40	PCR, ELISA IgM, MAT	[86]
**Typhus**								
Taiwan	1997	Retropective study	86	9	10.5	Mean: 38	ELISA IgM, MAT, IFA	[115]
China	2011	Case report	1	1	100	53	ELISA IgM, MAT	[116]
India	NM	Case report	1	1	100	40	ELISA IgM, MAT	[117]
India	NM	Case report	1	1	100	9	ELISA IgM, ICT	[118]
India	2015	Cross sectional study	258	10	3.88	NM	ELISA IgM, PCR	[119]
Tamil Nadu	2014–2015	Retrospective study	354	23	6.5	Mean: 31.48	ELISA IgM, MAT	[120]
**Typhus**								
India	2018	Retrospective study	22	9	41	Mean: 38	ELISA IgM, PCR	[121]
India	2017	Retrospective study	7	2	18	2–90	ELISA IgM	[122]
India	2018–2019	Retropective study	608	11	42.31	NA	ELISA IGM	[123]

Note: N, Sample size, ELISA—Enzyme linked immunosorbent assay, PCR—Polymerase Chain reaction, IgM—Immunoglobulin M, MAT—Microscopic Agglutination Test, DFM—Dark Field Microscopy, RMAT—Rapid malaria antigen test, qPCR—Quantitative Polymerase Chain reaction, RT-PCR—Reverse Transcription Polymerase Chain reaction, BFMP—Blood Film for Microscopy Parasite, NA—not available.

### 4.2. Malaria

Malaria is a vector borne disease, transmitted by female *Anopheles* mosquitoes, with approximately 229 million cases and 435,000 deaths. Malaria epidemics can occur when climate and other conditions favour malaria transmission in areas where people have little or no immunity. They can also occur when people with low immunity travel to areas with high malaria transmission, for instance to find work, or as refugees. Symptoms of malaria range from asymptomatic or uncomplicated such as fever, headache, myalgia, and general malaise to severe complications.

Malaria and leptospirosis are significant global infections with overlapping geographic distribution, especially in tropical and subtropical countries. As malaria and leptospirosis are common in the tropics, co-infection is to be expected and is more likely to occur by chance. However, there is a common practice in a malaria endemic that if an acutely febrile patient is found to be malaria-positive, malaria is assumed to be the sole cause of fever. Fever, chills, body aches, yellow urine, jaundice, vomiting, and headache were all significant symptoms for coinfection patients [107,108,109,110,111,112]. Thus, the clinical symptom of malaria and leptospirosis were similar, making accurate diagnosis difficult without any laboratory confirmation [95]. Failure to recognize acute leptospirosis co-infection may cause a delay in initiating proper therapy and leading to severe complications of leptospirosis such as hepatic and renal failure [110].

The effect of age group, sex, socioeconomic background and living status was associated with poor housing areas, lives in or travels to an area with risk of malaria and leptospirosis. The high proportion of coinfection among male patients was also associated with their job description. However, there is no significant difference between the age of male and female patients. Furthermore, the clinician may also underdiagnose or fail to recognize the infection due to the similarity in clinical presentation resulting in late treatment.

Even though coinfection of leptospirosis and malaria has rarely been reported, there should have a high index of clinical suspicion and, for those who present with fever, thrombocytopenia, multiorgan failure, empirical treatment (3rd generation cephalosporin and doxycycline along with antimalarial) should cover most of the causative agents in case of coinfection. If diagnostic facilities for leptospirosis are not available, it is beneficial to treat the co-infection with a combination of Doxycycline and antimalarial treatment.

Artesunate is used to treat severe malaria cases, whereas Ceftriaxone is effective against bacterial infections and doxycycline is an appropriate therapy against atypical organisms such as *Leptopsira* and rickettsia. Critically ill patients with severe malaria should be given third generation cephalosporins, Chloroquine or Quinine with doxycycline for effective treatment of both infections simultaneously.

### 4.3. Melioidosis

*Bukholderia pseudomallei* is a soil and water surface bacterium found in tropical areas that causes melioidosis after exposure to contaminated water or soils [114]. Melioidosis has been reported in Malaysia and other endemic countries such as Brazil [124], Australia [125], India [126] and Thailand [127]. Infection with *B. pseudomallei* is most commonly associated with an inoculating injury, ingestion, or inhalation of aerosolized bacteria and occurs more frequently in the wet season or following extreme weather events such as tropical storms.

Co-occurrence of leptospirosis and melioidosis has rarely been reported. Based on our extensive literature review, there are only three publications on melioidosis and leptospirosis coinfection which may be attributed to a lack of awareness and underreporting. Clinical presentations are known to overlap fever, jaundice, headache, myalgia, diarrhoea, cough and vomiting. Furthermore, the relative risk of melioidosis in diabetic patients is strong leading to death. 83% (5/6) of the patients reported from the three publications died due to the misdiagnosis or delayed diagnosis [86,113,114], When the leptospira and melioidosis coinfection was recognized, the surviving patient was closely monitored and administered with appropriate antibiotics until his condition improved. His fever was completely recovered in 10 days and was discharged with oral doxycycline and co-trimoxazole for five months.

*B. pseudomallei* is commonly susceptible to ceftazidime, amoxycillin-clavulanic acid, penicillin, imipenem, azlocillin, doxycycline, aztreonam and ceftriaxone but resistant to gentamicin and colistin [128]. Intravenous ceftazidime or meropenem is the preferred choice for initial therapy for most patients with melioidosis. Moreover, most of the patients were treated with IV ceftriaxone and doxycycline during the initial phase and increased the dose to ceftazidime and cystaline Penicillin after the confirmation of the diseases.

### 4.4. Typhus

Scrub typhus is a rickettsial zoonosis caused by *Orientia tsutsugamushi* and transmitted through bites by infected chiggers. Fleas, ticks, louse or mites become carriers of the bacteria when they feed on the blood of infected humans (epidemic typhus) or infected rodents (a reservoir of *Leptospira*). Typhus and leptospirosis are likely to share similar routes of transmission when rats are abundant. Humans usually become infected when infected flea feces contaminates excoriated skin or are inhaled [129]. Other studies have found that *R. rattus* and *R. norveigus* are the primary host of *R. typhi* infected *X. cheopsis* fleas in Indonesia, as well as maintenance hosts for *Leptospira* spp. [129,130,131]. Incidence of leptospirosis increases after the rainy season due to water logging resulting in contact with animal urine, similar to scrub typhus that increases due to the large trombiculid mite population [115].

Generally, patients may present with clinical features such as headache, myalgia, cough, abdominal pain, rash, thrombocytopenia, leukocytosis and nausea. It is difficult to differentiate between leptospirosis and scrub typhus infection. Most of the coinfected patients have higher median platelet counts with lower blood bilirubin and creatine concentration compared to leptospirosis patients alone. It is important that the clinician kept in their mind to performed other appropriate diagnosis if the patient is travelling to or returning from any endemic areas. In serious situations, the coinfection cases can be fatal if not treated appropriately. In one example, a patient presented with leptospirosis and was treated appropriately with high doses of penicillin. However, the patient’s condition become worse and died from respiratory distress syndrome which is one of the common causes of death from *O. tsutsugamuchi* infections [115,132]. Intravenous penicillin is one of the choice treatment on severe leptospirosis, however *O. tsutsugamushi* was resistant to these antibiotics [133].

Hence, timely detection and appropriate management of leptospira co-infection with *O. tsutsugamushi* is important to significantly reduce the severity of the cases. An early detection of leptospirosis and rickettsioses was presumed when the 53 old healthy fruit salemans presented severe sepsis with impending respiratory failure. Empiric antibiotics with intravenous penicillin (treatment of leptospirosis) and levofloxacin (treatment of rickettsioses and other negative bacterial infection) were given and the patient’s condition improved and urine output increased gradually. The patient was discharged after 10 days in hospitals and was asymptomatic at 1 month follow up [116]. Tetracycline also been used to treat patients with leptospirosis and scrub typhus. Even though the doxycycline has usually been used for mild leptospirosis, but in the serious situation, high dose penicillin remains the choice. Other like ceftriaxone, ampicillin and cefotaxime can be considered as alternatives for penicillin [117].

## 5. High Potential for Coinfection with Environmental Bacteria

There is a unique interaction between the host and the resident microbiome by providing innate immunity, metabolism, diseases, and nutrition [134]. Microbial cells can quickly adapt to environmental differences through a wide range of genotypic and phenotypic properties. This adaptation is crucial for bacterial survival under extreme conditions. Therefore, bacteria play a crucial role in ecosystem cycles such as soil structure improvement, water recycling, soil aggregation, and soil nutrient sequence [135]. Various pathogenic bacteria may be shed between the pores of soil aggregates and embedded in the complex structure of clay. The interaction of coinfection between environment, vector and human was shown in Figure 1. However, the survival of the bacterial community is also influenced by pH, type of soil, nutrients, climate change and crop type. To date, environment bacterial pathogens such as *Clostridium* [136], *Escherichia coli* [137], and *Bacillus* [114] have contributed significantly to human health. Although rarely reported, the possibility of coinfection of these bacterial pathogens and *Leptospira* might occur because they are easily found in water, soil, and contaminated foods. To date, there is a novel case reported in a 32-year-old Malaysian coinfected with *Leptospira* and *E. coli* during her post-partum period. The patient suffered from acute neurological deterioration, pulmonary haemorrhage, disseminated intravascular coagulopathy, and multi-organ failure [138]. The observed symptoms mirrored clinical presentations typical of headaches, acute fever, rash, jaundice, malaise, myalgia, and lethargy [32]. Further investigations should address the probability that humans may be directly or indirectly exposed to multiple local infections [139].

The persistence of *E. coli* in the environment is becoming a major issue for farmers and public health. Most outbreaks were triggered by the consumption of undercooked beef and tainted raw vegetables [140], decomposed manure, human sewage, slaughterhouse wastes, and animal slurry. The bacteria can live under favourable conditions for up to a year. For example, Zhang et al. [141] found that *E. coli* O157: H7 could live for 33 days in neutral soils and 7 days in acid soils. They also reported that the relationship between soil pH and organic carbon could increase the period of *E. coli* survival time but negatively correlated with exchangeable K. Shiga-like toxin *E. coli* (STEC) produces two different toxins (Stx 1 and Stx 2) and causes haemorrhagic colitis [142], haemolytic-uremic syndrome [143], and intestinal pathogenic *E. coli*. Once infected, *E. coli* may persist for up to one year; however, the resident strain shift in the human intestine is not well understood [144].

*Clostridium* encompasses more than 200 species of Gram-positive bacteria, with 50 species contributing to intestinal diseases in humans and animals [145]. It prefers growth under low oxygen levels and ideal conditions that can multiply rapidly. *Clostridium perfringens* can produce more than 20 virulent toxins and be clinically associated with systemic and enteric diseases, including food poisoning, enterocolitis, gas gangrene, fever, muscle tissue destruction, and massive local oedema. The first profiling of co-regulated bacterial transcriptomes showed that TLR2 and NLRP3 inflammasome genes were induced by 33% during clostridial myonecrosis infections, with the *C. perfringens* gene successfully regulated [146].

Bacteria belonging to the genus *Bacillus* are Gram-positive, rod-shaped, spore-forming, motile and easily found in the soil. Although used to protect plants and animals from microbial infection by secretion of antibiotics, volatile compounds, and enzymes [147], it was not considered pathogenic to humans, except for a few strains such as *Bacillus anthracis* that can cause anthrax in humans and animals. Humans usually get infected during contact with infected animals or spore ingestion due to the inhalation of spores from soil management activities [148]. Amazingly, the bacteria are highly resistant to desiccation and remained inactive for a few years until the condition is ideal. Anthrax is also a potential biological weapon in a biological attack or bioterrorism [149].

## 6. Future Direction

Based on the frequency of coinfections, it is vital to enhance the clinical examination, protocols, and interpretation of laboratory results to enable accurate clinical diagnosis. During arbovirus-related outbreaks, a systematic diagnostic protocol should include *Leptospira* as a possible co-infecting agent [150], and clinical practitioners should be made aware of the possibilities of coinfection and seek further consultation with infectious disease physicians before the patient faces severe complications [151].

Several questions remain to be answered, such as it is unclear whether the presence of multiple infections in patients may present as short- or long-term clinical presentations. There is a gap describing the possibilities of leptospirosis coinfection transmission. As coinfection of leptospirosis with other diseases is now more frequently reported, early diagnosis of leptospirosis coinfection requires clinical awareness, and appropriate treatment to prevent any disease severity and complications.

On the other hand, it is difficult to control the emergence of new bacterial coinfections with leptospira, particularly bacteria which reside in various parts of soil and water. Unravelling the unique interaction between leptospira and other pathogenic bacteria may increase the knowledge of new potential transmission. Shotgun Metagenomic DNA sequencing allows the analysis of the taxonomic structure and functional genetic capacity of the microbiome community [152]. Microbiome research has gained tremendous attention over the last decade, and it has become evident that microbiota associated with other organisms is important to support the health, development, and well-being of their hosts [153]. However, the advancement of genomics practices to clinical microbiology has taken a long and complicated way [154].

## 7. Conclusions

In conclusion, leptospirosis co-occurrence with other diseases is not often reported but is highly suspected, thus warrants further actions. As such, potential coinfection of *Leptospira* with other pathogens should be treated accordingly in high-risk patients. Medical practitioners may misdiagnose possible coinfections and should be aware of the appropriate diagnosis protocols prior to treatment. Although *Leptospira* infection occurs when humans are exposed directly to the infected animal and contaminated environment, there is a significant gap regarding the transmission of other pathogens and coinfection in leptospirosis patients. Further studies should focus on understanding the ecological aspect of the host and the persistence of *Leptospira* spp. with other pathogenic bacteria in the environment to provide optimal health control and prevention outcomes. Moreover, the development of a leptospirosis coinfection database from metagenom analysis could assist in the development of improved diagnostic and treatment of patients.

## Figures and Tables

**Figure 1 ijerph-18-09411-f001:**
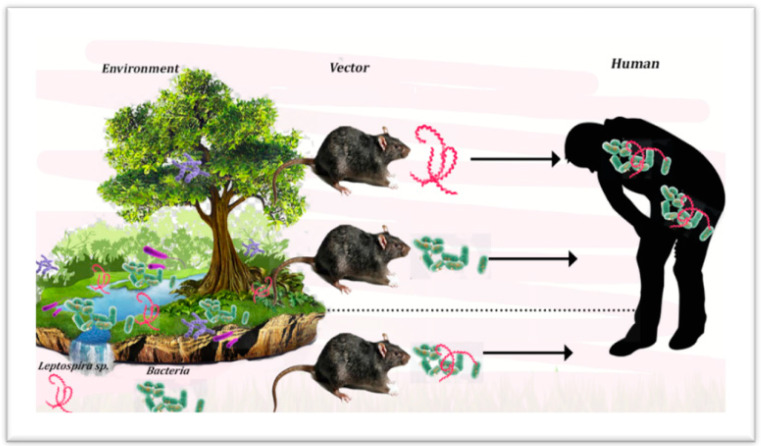
Environment and vector infections occur as a result of coinfections in human.

**Table 1 ijerph-18-09411-t001:** Example of pathogenic, intermediate and saprophytic *Leptospiral* species.

Genotypic Classification	*Leptospira* Species
Pathogenic *Leptospira*	*L. alexanderi*, *L. alstonii*, *L. borgpetersenii*, *L. interrogans*, *L. kirschneri*, *L. kmetyi*, *L. mayottensis*, *L. noguchii*, *L. santarosai*, *L. weilii*.
Intermediate *Leptospira*	*L. broomii*, *L. fainei*, *L. inadai*, *L. licerasiae*, *L. wolffii*, *L. venezelensis*, *L. broomii.*
Sapropytic *Leptospira*	*L. biflexa*, *L. meyeri*, *L. terpstrae*, *L. vanthielii*, *L. wolbachii*, *L. yanagawae*, *L. idonii.*

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
