# Peer review of "Leptospirosis and Coinfection: Should We Be Concerned?"

_ijerph, 2021, doi:10.3390/ijerph18179411_

Round 1

Reviewer 1 Report

This study is accurately described, with a particular eye on the enviromental factors which would be involved in the leptospiral-bacterial o viral coinfection.

I was wondering if it would be possible to describe a potential lepto-coinfection high risk patient (age, sex, general health condition and/or pathologies, medical treatments (immmune-depression, immune-modulation),..

I would check for little typing or grammar mistakes

Author Response

Reviewer # 1

General Comments:

This study is accurately described, with a particular eye on the environmental factors which would be involved in the leptospiral-bacterial o viral coinfection.

I was wondering if it would be possible to describe a potential lepto-coinfection high risk patient (age, sex, general health condition and/or pathologies, medical treatments (immmune-depression, immune-modulation).

Response 1: We want to thank the reviewer for their constructive comment and suggestion on this manuscript, it helped us to improve the manuscript. We have made substantial changes in several part of the paper to address the reviewer comments especially in each disease such as dengue (Page 5, Lines 225 – 243), Malaria (Page 8, Lines 264-300), Melioidosis (Page 8, Lines 301- 326) and Thypus (Page 9, Lines 327-363)

Reviewer 2 Report

Although the issue of co-infections and Leptospira seems interesting, the first 5 pages do not address co-infections at all. Instead they give a very general and high-level oversight of other aspects of the life cycle or epidemiology of the pathogen. I would strongly suggest reducing the first 5 sections of the review to a single section, hitting only the most pertinent and well-supported points. There is also a lot of redundancy throughout the MS, as well as many broad, unfounded (or outdated) statements. This review could use substantially restructuring, reducing, and synthesizing.

Section 6 seems completely unrelated to Leptospira, and the authors do not provide any rationale or linkage between leptospirosis and these other pathogens. This section could be removed.

Until Section 8, the authors do not directly address co-infections and Leptospira, and as such I think the title and abstract are misleading. Really, this is a broad overview of the factors related to environmental transmission of Lepto. That being said, Sections 8 and 9 were interesting, and I think the authors should re-focus their review on illuminating and understanding more about the instances of co-infection that have been documented, and trying to account for the source of these infections through the literature, rather than starting with a broad (and lengthy) description of Leptospira and leptospirosis in general.

I have also made extensive comments directly on the PDF.

Author Response

Reviewer #2

General Comments:

Point 1: Although the issue of co-infections and Leptospira seems interesting, the first 5 pages do not address co-infections at all. Instead, they give a very general and high-level oversight of other aspects of the life cycle or epidemiology of the pathogen. I would strongly suggest reducing the first 5 sections of the review to a single section, hitting only the most pertinent and well-supported points. There is also a lot of redundancy throughout the MS, as well as many broad, unfounded (or outdated) statements. This review could use substantially restructuring, reducing, and synthesizing.

Response 1: We thank the reviewer for his/her interest in our work and for his/her precious suggestion. We did our very best to revise the manuscript according to your insight and suggestions. We understand that some of the sections may seem to go into a lot of detail, but we consider this detail to be important if we are to really consider all aspects. We have made substantial changes in several part of the paper to address the reviewer comments.

Point 2: Section 6 seems completely unrelated to Leptospira, and the authors do not provide any rationale or linkage between leptospirosis and these other pathogens. This section could be removed.

Response 2: We thank the reviewer for his/her interest in our work and for his/her precious suggestion. In this section we describe about environmental bacteria pathogen that contributed to human health. However, there is a possibility of coinfection to be occur between these pathogen bacteria with leptospira spp. since they are easily found in water and soils. We have changed the title to “high potential coinfection from environmental bacteria”. We decided to maintain the information as the information is valuable to be use in other diagnosis probabilities (Page 9, Lines 367-415).

Point 3: Until Section 8, the authors do not directly address co-infections and Leptospira, and as such I think the title and abstract are misleading. Really, this is a broad overview of the factors related to environmental transmission of Lepto. That being said, Sections 8 and 9 were interesting, and I think the authors should re-focus their review on illuminating and understanding more about the instances of co-infection that have been documented, and trying to account for the source of these infections through the literature, rather than starting with a broad (and lengthy) description of Leptospira and leptospirosis in general.

Response 3: Thanks for your careful reading of our manuscript. We have revised and expanded the section of the section 8 and 9 and the detail is now included as advised. We also made substantial changes in several part of the paper to address the reviewer comments.

Specific Comments:

Point 1: “continue to rely on their interactions with other microorganisms”. I think this is a significant overstatement. Ref 20 identifies the presence of other microorganisms in environments with Lepto, but there's no evidence that Lepto RELIES on their presence. Although papers have shown that Lepto can form biofilms with other bacteria, it has also been shown to form biofilms alone.

Response 1: Thanks for your careful reading of our manuscript. We have revised and expanded the paragraph as advised. Page 2, Lines 54-56

“Although pathogenic leptospira can produce biofilm in nature [20-22], the interaction with other microorganisms in the environment might support the survival and persistence of leptospiral pathogens outside the host [23-24]”.

Point 2: “other pathogens may provide an optimal environment for the development of other pathogens”. The authors provide no evidence for the presence of other pathogens coexisting with Lepto. They only reference the one paper where other environmental microbes were found. This is a problematic overstatement.

Response 1: Thank you for your kind suggestion. As you suggested, we have deleted and replaced with new sentences. Page 2, Lines 57-63.

“Coinfection can influence epidemiology and disease severity [25]. A coinfection involving simultaneous infection of several pathogens in the same host may result in different types of diseases [26]. A coinfection is synonymous with simultaneous infection, mixed infection, multiple infections, concomitant infection, concurrent infection, polyinfection, polyparasitism, and multiple parasitism of an individual host [27-28]. Coinfection involves infectious pathogens from various taxa levels (bacteria, fungi, parasites, and virus) and genetic variations of the same infectious agents [29]”.

Point 3: “pathogens”. There is an important difference between microbes and pathogens.

Response 3: Thank you for your kind suggestion. As you suggested, we have changed it to the pathogenic leptospira as advised. Page 2, Lines 76.

Point 4: “To date, nine pathogenic species and six saprophytic Leptospira spp. have been identified by genetic analysis, while five Leptospira species exhibit intermediate pathogenic properties”. This is out of date, there are more species described now.

Response 4: Thank for your comments, we have included that information in the Table 1. Page 3. Line 106

Point 5: “It remains one of the most important public health issues due to the wide range of serovars circulating within large reservoirs”. More detail is needed here to support this statement.

Response 5: Thank you for your kind suggestion. As you suggested, we have rephased the sentence accordingly. Page 3, Lines 109-117.

“It remains one of the most important public health issues due to the wide range of serovars circulating within large reservoirs. Large populations of rodents, farm animals and dogs are capable of being colonised within the kidneys and sheding of Lepospira in urine which which increase the spread to the environment [51-53]. If Leptospira that persist in the renal and urinary systems are shed from animals, the bacteria can survive under optimum environmental conditions for months and up to a year [54-55]. Animals that have recovered from leptospirosis may continue to be carriers of Leptospira within the renal tubules for several months [56]”.

Point 6: “In addition, tropical climate factors can influence Leptospira infectivity by providing ideal conditions, with transmissions worsening during heavy rainfall and flooding. This is not surprising, as during flooding, most animals would naturally seek out higher ground”. Recent work from Goarant's group suggests that transmission worsens during floods b/c rain water flushes the spirochete from soil into the water system, promoting transmission. Please include all relevant detail when making these types of claims. Over simplification is an issue throughout this manuscript thus far.

Response 6: Thank you for your kind suggestion. As you suggested, we have rearranged the sentences and added the information as suggested. Page 3. Lines 118-127.

“Additionally, climate factors may influence Leptospira infectivity by providing ideal conditions for Leptospira to survive for prolonged periods of time, with transmissions becoming worse during heavy rainfall and flooding. Leptospira can easily contaminate the environment and pose a greater risk of infection through skin wounds during heavy rain. The water hits land surfaces and its either stored in the subsurface as soil moisture, or lost to evapotranspiration into freshwater bodies which leads to human to be be exposed [57-59]. This situation increases the possibility of Leptospira infections among rodents and increased interaction with human will lead to increased occurrence of leptospirosis [60]. Reports on flooding is a significant factor for the increase in leptospirosis cases, and are common in Thailand, the Philippines, Sri Lanka, and Malaysia [61-64]”.

Point 7: “weeks or months”. Studies have shown even longer environmental survival, also a reference is required here.

Response 7: Thank for your comments, we added the citation. Page 3, Lines 113-115.

“If Leptospira that persist in the renal and urinary systems are shed from animals, the bacteria can survive under optimum environmental conditions for months and up to a year [54-55]”.

Point 8: “Occupational groups such as banana plantation farmers, sugarcane workers, paddy farmers, veterinarians, animal shelter employees, hunters and abattoir workers are at high risk of infection [33]. Indirect contact with the environment, such as water and soil, are common and widely associated with the recreational, holiday, and occupational activities [34-35]”. These facts were also stated earlier and this section seems a bit random. Should be tightened and reworked.

Response 8: Thank you. We have revised the paragraphs according to suggestion. Page 4, Lines 138-147

“Occupational groups such as banana plantation farmers, sugarcane workers, paddy farmers, veterinarians, animal shelter employees, hunters, wet market workers and abattoir workers are at high risk of infection due to their job description involves contact with water and soils [69-70]. They are typically prone to skin cuts, abrasions during the activities in the particular areas which are providing a suitable environment and rich with source of food favouring the presence of rodents [71]. The rodent might shared similar places with people, easily moving around and excreting bacteria into environment via urine [72]. Farm labourers are especially vulnerable to injury, increasing their susceptibility to bacterial infections, partially due to a lack of adherence to the need for personal protective equipment”.

Point 9: “Leptospira Transmission from Soil or Water”. This whole section needs to be tightened and synthesized. At the moment it reads like the authors have summarized each paper in a sentence, rather than pull the information together and relay the overall meaning(s). There is also too much repetition with earlier sections.

Response 9: We have revised the section by adding and deleting some sentences as suggested. Page 4, Lines 148-182.

Point 10: “Routinely”. There is only one reference, yet the authors state that this is routine. Please add additional references here

Response 10: We have added new references as suggested [73-76].

Point 11: “To date, a few bacterial pathogens have contributed significantly to human health, such as Clostridium [65], Escherichia coli [66], and Bacillus [67]. The bacterial infection was rarely reported, but the possibilities of coinfection between these bacterial pathogens with Leptospira might occur because they are easily found in water, soil, and other foods heeded unsafely”. The first part of this is a gross understatement - perhaps the authors meant environmental pathogens? The second part is really just a hypothesis, and the authors do not support it with evidence.

Response 11: We are appreciated the reviewer comment. Yes, the first part refers to the pathogenic bacteria found in the environment. In the second part, we are attempting to explain the possibility of other environmental pathogenic bacteria coexisting-alongside Leptopsira. Because a novel finding of leptospirosis coinfection with E. coli was discovered, there should be a high possibility for transmission between these other pathogens to be coinfection with leptospira due to the similar environmental conditions. Page 10, Lines 376-387.

“To date, environment bacterial pathogens such as Clostridium [138], Escherichia coli [139], and Bacillus [115] have contributed significantly to human health. Although rarely reported, the possibility of coinfection of these bacterial pathogens and Leptospira might occur because they are easily found in water, soil, and contaminated foods. To date there are novel case reported in a 32-year-old Malaysian coinfected with Leptospira and E. coli during her post-partum period. The patient suffered from acute neurological deterioration, pulmonary haemorrhage, disseminated intravascular coagulopathy, and multi-organ failure [140]. The observed symptoms mirrored clinical presentations typical of headaches, acute fever, rash, jaundice, malaise, myalgia, and lethargy [141]. Further investigations should address the probability that humans may be directly or indirectly exposed to multiple local infections [142]”.

Point 12: “79-80”. These references are unrelated to Lepto and this statement seems purely speculative. Suggest to remove.

Response 12: Thank you for your kind suggestion. As you suggested, we have deleted this statement.

Point 13: “Leptospirosis coinfection is a special case and also known as “the great mimicker”. This statement should be referenced.

Response 13: Thank you. We have added references as suggested. Pages 4-5, Lines 187-189

“Leptospirosis coinfection with other pathogenic bacteria in humans is a special case and also known as “the great mimicker” [35 ,89-90]”.

Point 14: “The clinical presentation of leptospirosis coinfection is considerably overlapping, leading to misdiagnosis and mistreatment. Patients may experience acute febrile illness in the acute stage and may suffer long-term infections or deaths”. Would suggest looking at this paper: https://journals.plos.org/plosntds/article?id=10.1371/journal.pntd.0007205.

Response 14: Thank you very much for bringing to our attention the works of Dr. Smith et al. (2019) and the suggested reference is cited in this paragraph. Page 5, Lines 190-193.

“The clinical presentations of leptospirosis coinfection are considerably overlapping, leading to misinterpretation and maltreatment. Patients may develop fever, abdominal pain, myalgia, headache, vomitting, as well as long-term infections or death [35-36, 91-92]”.

Point 15: “Therefore, bacterial coinfection in leptospirosis is crucial to determine treatment appropriately and warrant further investigation”. I think it might be worth noting that doxycycline is the most common treatment for Lepto, and that it is broadly effective against other pathogens and is not associated with AMR. In this context, the authors should discuss the likelihood that treatment with doxy (or other things) would or would not change, depending on the co-infection present.

Response 15: Thank you for your kind suggestion. As you suggested, we have rephased the sentence accordingly by adding the information of treatment as suggested. Page 5, Lines 194-203.

“The majority of leptospirosis cases are mild and resolve spontaneously. Minor cases of leptospirosis resolve with time and oral antibiotics such as doxycycline, azithromycin, ampicilin and amoxicilin, which are administered based on the severity of the illness. Treatment with antibiotics must be initiated as soon as provisional diagnosis of leptospirosis is suspected, regardless of the length of the symptom [93]. Furthermore, most of the coinfected patients respond effectively to fluid therapy, doxycycline and closely monitoring of their platelet count and hematocrit [94-95]. However, in some leptospirosis coinfections, treatment could not be assessed due to the lack of data, and it is critical to determine these different of infections that can provide a relevant treatment dicisions for patients with coinfections [96]”.

Point 16: “elevated liver transmission leading to death”. All co-infected patients died?

Response 16: As suggested, we have revised and delete the not appropriate statement.

Point 17: “Several studies have shown that pathogenic bacteria shed together with leptospira caused coinfection diseases”. This should be expanded upon and referenced.

Response 17: We deleted the paragraph as we found that it was not appropriate for this section.